# Socioeconomic position, social mobility, and health selection effects on allostatic load in the United States

**Alexi Gugushvili**[1,2,3]*, **Grzegorz Bulczak**[2], **Olga Zelinska**[2], **Jonathan Koltai**[4]

**1** Deparament of Sociology and Human Geography, University of Oslo, Oslo, Norway, **2** Institute of Philosophy and Sociology, Polish Academy of Sciences, Warsaw, Poland, **3** Nuffield College, University of Oxford, Oxford, United Kingdom, **4** Department of Sociology, University of New Hampshire, Durham, New Hampshire, United States of America

\* alexi.gugushvili@sosgeo.uio.no

**Data Availability Statement:** The data underlying the results presented in the study are available from Add Health https://addhealth.cpc.unc.edu/data/.

## Abstract

The contemporaneous association between higher socioeconomic position and better health is well established. Life course research has also demonstrated a lasting effect of childhood socioeconomic conditions on adult health and well-being. Yet, little is known about the separate health effects of intergenerational mobility—moving into a different socioeconomic position than one's parents—among early adults in the United States. Most studies on the health implications of mobility rely on cross-sectional datasets, which makes it impossible to differentiate between health selection and social causation effects. In addition, understanding the effects of social mobility on health at a relatively young age has been hampered by the paucity of health measures that reliably predict disease onset. Analysing 4,713 respondents aged 25 to 32 from the National Longitudinal Study of Adolescent Health's Waves I and IV, we use diagonal reference models to separately identify the effects of socioeconomic origin and destination, as well as social mobility on allostatic load among individuals in the United States. Using a combined measure of educational and occupational attainment, and accounting for individuals' initial health, we demonstrate that in addition to health gradient among the socially immobile, individuals' socioeconomic origin and destination are equally important for multi-system physiological dysregulation. Short-range upward mobility also has a positive and significant association with health. After mitigating health selection concerns in our observational data, this effect is observed only among those reporting poor health before experiencing social mobility. Our findings move towards the reconciliation of two theoretical perspectives, confirming the positive effect of upward mobility as predicted by the "rags to riches" perspective, while not contradicting potential costs associated with more extensive upward mobility experiences as predicted by the dissociative thesis.

**Funding:** This work was supported by the Polish National Science Centre grant received by AG (Program SONATA14) - https://ncn.gov.pl/ - [grant number UMO-2018/31/D/HS6/ 01877]. The funders had no role in study design, data collection and analysis, decision to publish, or preparation of the manuscript.

**Competing interests:** The authors have declared that no competing interests exist.

# 1. Introduction

Socioeconomic position is a fundamental cause of health disparities [1]. Those occupying higher rungs on the socioeconomic ladder tend to experience lower rates of morbidity and mortality compared to those placed lower in the social hierarchy [2,3]. In addition to socioeconomic position attained in adulthood, socioeconomic origins exert significant and independent effects on later life health [4,5], reflecting the "long arm" of childhood circumstances [6–8]. The enduring effects of childhood circumstances are thought to represent the downstream consequences of cumulative advantage and disadvantage, whereby stresses and strains accrue over the life course to a greater extent among those in socially disadvantaged positions, setting in motion more rapid aging or weathering of biological systems under conditions of chronic adversity [9,10].

An unresolved question in social stratification and social epidemiological research is whether the movement between origin and destination socioeconomic positions, *per se*, influences health net of origin and destination effects. Because social mobility is linearly dependent on both social origin and destination, traditional regression frameworks are not able to separately estimate the effects of socioeconomic origin, destination, and social mobility simultaneously [11–13], calling into question much of the existing evidence [14]. While Sobel's diagonal reference models overcome this methodological limitation [15], few studies have utilized this statistical approach when investigating the health effects of social mobility, particularly in the North American context. This represents an important gap in the literature given renewed scholarly interest in intergenerational transmission of (dis)advantages and declining social mobility in the United States [16–18].

## 1.1. Key theories on health consequences of social mobility

Two main theoretical perspectives predict, respectively, negative and positive health consequences of upward social mobility. Sorokin's dissociative thesis views upward social mobility as a deviation from expected continuity associated with individuals' social origins [19]. Adjusting to an unfamiliar socioeconomic environment, while also socially distancing from the familiar and more natural past environment, can be a major stress-inducing process compromising upwardly mobile individuals' psychological and, consequently, physical health. In turn, an alternative perspective, so-called "rags to riches" thesis [20], suggests that upward social mobility could lead to better health outcomes by generating a sense of personal control, boosting psychological well-being from overcoming life course constraints, fostering healthy behaviours and lifestyles, and developing a health conducive sense of gratitude among the upwardly mobile individuals [21–25].

In recent years scholarly interest in health consequences of downward rather than upward social mobility has become particularly salient. This is in line with the post-liberal theory of social stratification which views social mobility primarily in terms of offspring attaining worse off living conditions than their parents did [26]. The so-called "falling from grace" thesis implies that downward social mobility leads to an undesirable loss of an ascribed socioeconomic position at birth and associated changes in practices, behaviours, and norms [27,28]. The perception of downward mobility as undeserved and unjust, together with the overall psychological maladjustment to a new environment, can precipitate chronic stress and thus compromise the health of downwardly mobile individuals [29,30]. Downward mobility may also increase the stress associated with financial hardship, a well-known correlate of physical and mental health [31]. Given these multiple perspectives, one of the goals of our study is to derive new evidence on the merits of the main health-related theories of social mobility.

## 1.2. Independent social mobility and health selection effects

An overview of the existing studies does not provide conclusive answers about social mobility effects on health, as many studies report null findings [14,32]. When significant associations of upward social mobility and health are identified, these links are not usually detrimental for health [20], while downward mobility in a number of studies was found to be damaging to health [33]. Inferring from these findings, however, is problematic because researchers use different indicators of mobility in socioeconomic position such as occupational class, status, education, and income. Moreover, social mobility research is characterised by at least two significant methodological constraints: first, relatively few studies use a statistical approach which is able to separately identify the relative importance of origin and destination socioeconomic position, while also isolating the effect of social mobility on health outcomes; and second, most studies rely on cross-sectional datasets, and are thus unable to differentiate between health selection and social causation effects when studying health implications of social mobility.

Referring to the first problem, many studies on the health consequences of social mobility continue to apply conventional regression approaches which usually differentiate social mobility trajectories by combining individuals' origin and destination positions and subsequently comparing health outcomes between these mobility groups; alternatively, some scholars simply omit from models either origin or destination socioeconomic positions to produce an estimate for health effect of upward or downward social mobility [14,34]. These analytical strategies are useful if researchers primarily intend to identify the role of origin and/or destination socioeconomic position for individuals' health, but they are unable to differentiate if health outcomes are independently affected by position and mobility effects. To mitigate this concern, researchers have proposed a special form of regression model that was developed to estimate the relative effects of two hierarchically ranked socioeconomic positions and the net effect of a movement between these two positions on the outcome variable of interest [11].

The second methodological concern implies addressing a likely bias stemming from health selection by which individuals' initial poor and good health not only, respectively, limits upward and facilitates downward mobility, but also is causally related to later life health. If health before social mobility is not accounted for, any possible health effects of social mobility can be erroneously attributed to social causation rather than to the health selection effects [35]. The role of health selection has been shown to be stronger for a transition process from adolescence to adulthood than for later life course transitions [36]. Considering the importance of initial health for educational attainment and resultant success on labour market, an intriguing and underexplored question is what are the health implications for those individuals who, regardless of initial health constraints, still manage to experience upward social mobility? Both positive and negative consequences can be predicted as mobility for disadvantaged individuals might imply greater costs, yet they might also derive greater psychological benefits from overcoming barriers on their social mobility journeys [37].

## 1.3. Social mobility and health measures

Adding to complexity, social mobility's effects might differ not only if initial health is accounted for, but also depending on the type of health measures used in studies. The most prevalent outcome variable employed by scholars, due to its wide availability in social surveys, is self-rated health [34,38,39]. Yet, self-rated health is not a perfect predictor of objective indicators of health and people might think of different aspects of wellbeing while assessing their health status [40,41]. Some studies also use depressive symptoms to identify possible effects of social mobility on health, but they are not able to capture any effects of social mobility on

physical health [20,42]. Timing of death, on the other hand, can be considered as a reliable indicator of health outcome, but data on mortality is mainly useful for studying later life health based on panel/cohort surveys or register-based datasets [43,44]. In this study, we use allostatic load, an index of multi-system physiological dysregulation among individuals [32], to identify the health implications of social mobility. This measure takes into account various aspects of health and provides information on valid variation in health already at a relatively young age [45,46].

## 1.4. Heterogeneous position and mobility effects

Existing studies on health inequalities suggest that different sociodemographic and socioeconomic groups have vastly different levels of allostatic load. Individuals' age, for instance, is strongly associated with allostatic load, which might imply that across individuals' life course mobility effects on AL also vary. The recent evidence also suggests that mobility effects across European societies are more pronounced among young people than among the elderly [39]. One of the explanations for this could be that psychological and stress-related costs and benefits of social mobility experience have primarily short-term effects that dissipate later in the life course [36]. Existing research on the health consequences of social mobility also indicates that the origin socioeconomic position might matter more for women than for men, while, in turn, mobility effects are more salient for men than women [20]. Social origin, attained socioeconomic position, and mobility between origin and destination positions might have varying implications for various sociodemographic groups which are known to have vastly different health and wellbeing outcomes due to historical, institutional, and structural differences. Race is of central importance in these respects.

In the United States, racial and ethnic inequalities in health and illness are well documented [47], and a growing body of research demonstrates the myriad ways that structural racism contributes to these disparities [48]. In addition to contemporaneous inequities, Gaydosh and colleagues argue that the health effects of social mobility may hinge on the disproportionate stressors experienced by racialized groups [49]. The John Henryism hypothesis, for example, posits that African American individuals are exposed to myriad psychosocial adversities throughout the life course, and that the active coping strategies employed to cope with such exigencies result in biological wear and tear [50,51]. Recent studies also suggest African Americans from disadvantaged backgrounds form a "skin-deep resilience," wherein a higher sense of control may lead to favourable psychological outcomes but greater physiological dysregulation [52–55].

## 1.5. The United States as a case study

The United States is an interesting case to study the health consequences of social mobility as it is characterized by widening socioeconomic inequalities [56] and declining levels of intergenerational social mobility [17,57]. The United States also has one of the most comprehensive panel datasets, described in detail below, which allows us to investigate socioeconomic origin and destination, social mobility, and selection effects on individuals' health. The same dataset used in the present study has been previously employed to investigate different health aspects of social mobility, including adolescent stressful experiences [58], early adversity on later life health through psychosocial resources [59], and the effects of life course socioeconomic position on cardiovascular health [60]. In turn, our contribution to the relevant scholarship is that we study consequences of social mobility on health by (1) constructing a robust indicator of socioeconomic position for both individuals and their parents; (2) identifying the relative importance of origin and destination socioeconomic positions on health; (3) detecting any

residual effects of social mobility; (4) testing if position and mobility effects differ by sociodemographic characteristics such as gender and race; and (5) examining the role of health selection in the observed associations.

## 2. Methods

### 2.1. Dataset

The National Longitudinal Study of Adolescent Health (Add Health) is a representative longitudinal study of individuals in the United States who were adolescents in the beginning of the 1990s. The study started in 1994–95 with Wave I of the panel which included data on 20,745 adolescents aged 12 to 19. By the time of writing, in the latest publicly available Wave IV, conducted in 2007–2008, the number of interviewed participants declined to 15,701 (76% of the original sample) with their average age of 29. Attrition did not occur completely at random, but rather the main identified predictors of response in Wave IV were individuals' gender, race, and parental education [61]. For this analysis, we use data from Waves I and IV. Due to cost-related considerations, in this study, we used the public-use version of Add Health with about 40% of respondents chosen randomly from the restricted-full sample. The main differences between the public and the restricted versions of Add Health arise due to confidentiality concerns. The full version of the dataset contains more sensitive information on respondents including their romantic relationships and DNA-related data. The later information is not of primary interest for our study, while the random mode of selection of participants for the public-use version of Add Health ensures that it is a representative survey data of the United States population of the relevant age. After list-wise deletion of observations with missing information, 4,713 individuals were available for our analysis.

### 2.2. Measures

**2.2.1. Health outcome.** Numerous past studies used the Add Health to investigate the impact of socioeconomic position on physical [62–64] and mental [49,55,65] health outcomes. Our goal was to construct a measure that would reliably detect health status determined by individuals' long-term socioeconomic conditions as well as their experienced stress levels from Add Health's Wave I to Wave IV. In this regard, one of the most appropriate indicators is individuals' allostatic load (further AL). AL identifies multidimensional physiological dysregulations that contribute to an onset of disease progression [66]. AL index may incorporate neuroendocrine, immune, metabolic, and cardiovascular system functioning and is a validated predictor of morbidity and mortality outcomes, especially at the earlier stages of life [67].

There are alternative approaches to construct AL and no consensus exists concerning which is the most appropriate method [68]. In this study, building on the previous research [32,69,70], we constructed AL index using biomarkers data from a blood test and medical examinations collected at Wave IV. Our AL index is based on seven biomarkers divided into five categories: (1) Lipid—Total to High-Density Lipoprotein Cholesterol; (2) Glucose—Glucose MG/DL; (3) Inflammation—C-reactive protein (CRP); (4) Body Mass Index (BMI); and (5) Cardiovascular–(5.1) systolic and (5.2) diastolic blood pressure and (5.3) resting heart rate. Our approach to constructing this measure is closely matched with the previous research in which AL is based on lipid and glucose metabolism, inflammation (C-reactive protein and fibrinogen), body fat deposition (body mass index and waist measurement) and cardiovascular measures [32]. We first separately z-transformed the described biomarkers and then estimated the mean score of these transformed biomarkers. Finally, we z-transformed the derived mean AL score. Another approach would be to flag the biomarkers if they are above a relevant medical threshold [71]. Our preferred measure, however, is more sensitive as it captures the full

variation in individuals' AL and therefore may help to identify individuals that will develop more serious health problems in the future. We consider this measure to be particularly appropriate for our study as it is able to capture even relatively small changes in young adults' health. This is especially important in the context of social mobility where the sensitivity of health outcome measures has yielded in mixed results [32,72]. For more detailed descriptive information about each employed component of AL index, refer to supporting information, S1 Table.

**2.2.2. Social origin, destination, and mobility variables.** For individuals' social origin variables, we utilised information on parental characteristics collected directly from parents at Wave I, while individuals' social destination variables are constructed from their attained socioeconomic position at Wave IV. A difference between origin and destination variables was classified as upward or downward social mobility.

Out of available measures of socioeconomic position in Add Health, we focused on individuals' and their parents' educational and occupational attainment which aligns with prior social stratification research in the United States [73]. Education is a known predictor of individuals' health [74] and since the average age of respondents at Wave IV is 29, most participants have completed their educational attainment [75]. Formal educational credentials, however, measured in the survey did not fully capture individuals' socioeconomic position due to unobserved heterogeneity in, among other areas, specific skills, productivity, and the quality and type of education, especially considering the fragmented and stratified educational system of the United States [76]. Therefore, we also utilised information on individuals' occupational attainment which is a validated proxy for their earnings, work autonomy, and job security [77]. Combining educational and occupational information allowed us to generate a robust measure of socioeconomic position with known links to health outcomes [78].

For parental education, we used the highest level of education obtained by parents [79]. For example, we rely on mothers' education if father completed high school and mother completed college. To construct our measures, we collapsed 10 educational categories for parents and 13 educational categories for respondents into 5 categories. The difference in the number of categories arose due to additional postgraduate degrees for respondents. We coded education variables as follows: some high school and lower (= 1), completed high school (= 2), some college (= 3), completed college 4-year degree (= 4), and completed some postgraduate qualifications (= 5).

Next, based on the previous research [80,81], we created occupational attainment variables for parents and respondents. In the case of parents, occupational data consisted of 10 occupational groups. We used the highest level of occupation obtained by the parents to construct five occupational categories. This was done by taking the average Nam-Power-Boyd scale score [82] for each of the 10 occupational groups and collapsing them into 5 categories from having no occupation (= 1) to Nam-Power-Boyd scale score from 70 to 100 (= 5). Individuals' occupational status was based on the Standard Occupational Classification codes converted into status scores based on the Nam-Power-Boyd scale. We created quintiles from the converted occupational status scores (the lowest status jobs = 1, the highest status jobs = 5).

Finally, to derive the index of socioeconomic position for parents and individuals, we combined educational and occupational attainment variables. This resulted in scores ranging from 2 to 10 points for the highest achieving individuals. To ensure that each mobility group had adequate representation, we collapsed the combined socioeconomic position scores into quintiles, where quintile 5 represents the top 20% (highest attainment based on educational and occupational status). From these combined measures we calculated intergenerational social mobility variables. We subtracted parental from individuals scores. This resulted in a mobility measure ranging from -4 to 4, where 0 represents the immobile group. For example, if the

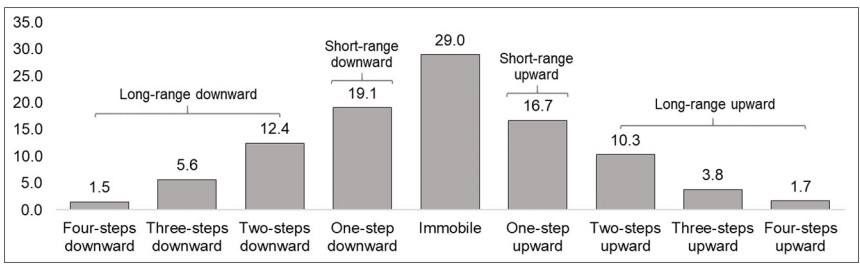

**Fig 1. Social mobility trajectories.** *Note*: Number of observations—4713.

respondent achieved a score equal 5, highest attainment (top quintile) and parental attainment was equal to 4, the difference between the two scores produces one-step upward mobility.

To ensure sufficient variation we collapse two, three and four steps into a long-range mobility indicator, separately for upward and downward mobility, while one-step mobility represents short-range mobility. Fig 1 shows the distribution of social mobility patterns with the immobile group being the largest single category of individuals, yet approximately 39% of respondents experienced downward mobility compared with 32% for upward mobility. In the empirical analysis, we differentiate between short-range (one-step) mobility and long-range mobility (two-four-steps).

**2.2.3. Confounders and initial health status.** In all multivariable models, we accounted for respondents' gender and age. Additionally, based on the previous research on predictors of health in the United States [83], in the main analyses we controlled for respondents' race/ethnicity (White [non-Hispanic], Black [non-Hispanic], Hispanic, and other), the type of residential area (rural = 1), and marital status (1 = married) [84].

To address possible health selection effects due to initial health status which could affect both social mobility experience and Wave IV health, we first utilised information on respondents' self-rated health at Wave I. More specifically, the respondents were asked the following question: "In general, how is your health?" which they could rate on from 1 (= poor) to 5 (= excellent) health. Second, we accounted for individuals' initial BMI scores. Third, we constructed the chronic disease indicator based on a set of four questions: "Do you have difficulty using your hands, arms, legs, or feet because of a permanent physical condition?"; "Do you have a permanent physical condition involving a heart problem?"; "Do you have a permanent physical condition involving asthma?"; "Do you have a permanent physical condition involving other breathing difficulties?". If the respondent answered yes to any of these questions the indicator equals 1 and 0 otherwise. Correlations between the outcome measure and selected Wave I health measures are presented in Supporting information, S2 Table. Although other Wave I health outcome variables were available in the dataset (e.g. health assessed by parents and depressive symptoms based on the Center for Epidemiological Studies Depression Scale (CES-D scale) [85]), additional analysis in Supporting information, S3 Table, shows that the selected health measures at Wave I are robust predictors of AL at Wave IV. Table 1 presents descriptive statistics for the described health outcome measure, confounding variables, and initial health status.

## 2.3. Statistical analysis

We used diagonal reference models (DRM) to identify associations between different socioeconomic positions and AL score, to assess the relative importance of origin and destination socioeconomic positions, and to estimate the consequences of social mobility as a deviation

**Table 1. Descriptive statistics.**

| | Mean | SD | Min | Max |
|---|---|---|---|---|
| AL at Wave IV | 0.00 | 1.00 | -2.47 | 6.65 |
| Respondents' age at Wave IV | 28.98 | 1.76 | 25.00 | 32.00 |
| Respondents gender (male = 1) | 0.47 | 0.50 | 0.00 | 1.00 |
| Race/ethnicity | | | | |
| White | 0.60 | 0.49 | 0.00 | 1.00 |
| Black | 0.24 | 0.43 | 0.00 | 1.00 |
| Hispanic | 0.11 | 0.32 | 0.00 | 1.00 |
| Other | 0.05 | 0.22 | 0.00 | 1.00 |
| Rural-urban divide (rural = 1) at Wave I | 0.29 | 0.45 | 0.00 | 1.00 |
| Marital status (married = 1) at Wave IV | 0.44 | 0.50 | 0.00 | 1.00 |
| Less than very good health at Wave I | 0.31 | 0.46 | 0.00 | 1.00 |
| BMI at Wave I | 22,29 | 4,38 | 11,12 | 55,94 |
| Chronic health condition Wave I | 0,02 | 0,15 | 0,00 | 1,00 |

*Note*: Number of observations– 4713.

from what could be expected from predicted health status of immobile individuals in origin and destination socioeconomic groups. DRM design allows overcoming multicollinearity problem arising due to social mobility measures being calculated directly from origin and destination socioeconomic position variables. In other words, conventional statistical models cannot simultaneously include origin, destination, and mobility parameters.

DRM is argued to be one of the most suitable methods for estimating an effect of social mobility because it disentangles this effect from the origin and destination position effects. An extensive overview of this statistical method, its usefulness in modelling of social mobility effects, and a comparison with conventional regression approaches are described and demonstrated elsewhere [11,14,29]. The key aspect of DRM is that immobile individuals' health is estimated by weighted mean values of AL for those who occupy diagonal cells in our two-dimensional five by five matrix (see Table 2 in Results' section). After accounting for immobile

**Table 2. Mean AL levels by parental and individuals' socioeconomic position.**

| Parental socioeconomic quintiles | Individuals' socioeconomic quintiles | | | | |
|---|---|---|---|---|---|
| | Lowest | Middle-low | Middle | Middle-high | Highest |
| Lowest | 0.20 | 0.04 | 0.08 | -0.15 | -0.01 |
| | [0.11;0.29] | [-0.09;0.18] | [-0.05;0.20] | [-0.34;0.03] | [-0.24;0.23] |
| Middle-low | 0.20 | 0.08 | -0.12 | 0.01 | -0.19 |
| | [0.09;0.31] | [-0.08;0.24] | [-0.24;0.00] | [-0.24;0.25] | [-0.37;-0.02] |
| Middle | 0.10 | 0.11 | 0.10 | 0.10 | -0.20 |
| | [-0.04;0.23] | [-0.07;0.29] | [-0.02;0.21] | [-0.19;0.39] | [-0.37;0.04] |
| Middle-high | -0.01 | 0.02 | 0.05 | -0.09 | -0.28 |
| | [-0.15;0.14] | [-0.14;0.18] | [-0.06;0.16] | [-0.23;0.04] | [-0.38;-0.18] |
| Highest | 0.10 | 0.03 | -0.06 | -0.18 | -0.32 |
| | [-0.11;0.31] | [-0.18;0.23] | [-0.19;0.06] | [-0.36;0.01] | [-0.42;-0.22] |

*Notes*: 95% confidence intervals in parentheses, number of observations– 4713. Quintiles are derived from combined educational and occupational attainment for parents and individuals respectively.

individuals' health, DRM estimates the relative strength of the effect of the origin socioeconomic position to that of own socioeconomic position and this so-called "weight" parameter takes values between 0 and 1. A higher value of this weight indicates a greater relative effect of destination characteristics on the outcome measure, AL score in our case. In the case when the weight parameter is equal to 0.5 it can be concluded that the origin and destination characteristics play an equally important role in determining AL.

To estimate possible effects of social mobility on AL in DRM, diagonal intercepts (values of outcome variable of immobile individuals) and weight parameter are jointly used to specify a cell-specific intercept for each off-diagonal cell (specific downward and upward mobility trajectories) in the two-dimensional mobility table. After predicting values for all off-diagonal cells, the DRM approach specifies the effect of social mobility over and above the value of AL conditioned by specific characteristics of origin and destination socioeconomic positions. Social mobility variables and associated point estimates, βs, in DRM approach could be interpreted in the same way as in a conventional regression model, a reference category being a group of individuals with the same socioeconomic position as their parents. To test if the effect of the social origin on AL and the health implications of social mobility varied by the individuals' socioeconomic position, their social mobility experiences, and other sociodemographic characteristics, in subsequent models we also derived point estimates for interaction terms between the weight parameters, social mobility, and confounding variables.

Lastly, as described in the Introduction section, we attempted to consider health selection of individuals into social mobility trajectories, which may also explain the later life health outcomes. For this purpose, we accounted for individuals' initial health at Add Health Wave I. We first controlled for initial health variables and then fitted DRMs separately for individuals with good and with poor initial health status. All DRM estimates are derived using "drm" module in Stata 16 statistical software [86]. For various empirical applications of DRM approach in different countries and contexts readers can refer to studies on, among other areas, redistribution preferences [87], likelihood of smoking [88], attitudes toward immigrants [89].

## 3. Results

### 3.1. Social gradient and descriptive mobility effects

Table 2 presents the mean levels of AL by individuals' origin and destination socioeconomic positions. The diagonal cells consist of AL score for intergenerationally immobile individuals, while the remaining cells above and below the diagonal show, respectively, the mean AL score among upwardly and downwardly mobile individuals. Expectedly, the mean AL score is higher for immobile individuals with low socioeconomic position (0.20, CI 0.11,0.29) in comparison to immobile individuals with a high socioeconomic position (-0.32, CI -0.42,-0.22). Off diagonal cells also suggest that the upwardly mobile individuals have lower AL levels. A similar but inverse pattern can be observed for those who experienced downward mobility. These associations could be partially explained by the fact that the upwardly mobile groups do not include those who ended up in the bottom quintile, while downwardly mobile groups do not include those who ended up in the highest quintile. The latter also suggests that upwardly and downwardly mobile individuals differ by their social origin and destination positions and to disentangle these position effects from social mobility effects, we employed the above-described statistical approach—DRM.

### 3.2. Origin and destination weights and multivariable mobility effects

Table 3 presents findings from multivariable DRM estimations. In all models, we accounted for individuals' age and gender, while in full models we also controlled for individuals' race/

**Table 3. Point estimates from DRM on AL levels.**

| | Model 1 | Model 2 | Model 3 | Model 4 | Model 5 |
|---|---|---|---|---|---|
| *Immobile socioeconomic quintiles* | | | | | |
| Lowest | 0.20*** | 0.19*** | 0.17*** | 0.19*** | 0.17*** |
| | [0.13,0.26] | [0.13,0.26] | [0.10,0.24] | [0.12,0.26] | [0.09,0.24] |
| Middle-low | 0.070 | 0.07 | 0.07 | 0.07 | 0.07 |
| | [-0.01,0.15] | [-0.02,0.15] | [-0.01,0.15] | [-0.02,0.16] | [-0.02,0.15] |
| Middle | 0.08* | 0.09* | 0.11** | 0.09* | 0.11** |
| | [0.01,0.15] | [0.02,0.17] | [0.04,0.19] | [0.01,0.17] | [0.04,0.19] |
| Middle-high | -0.08 | -0.09* | -0.11** | -0.09* | -0.10* |
| | [-0.16,0.01] | [-0.17,-0.02] | [-0.18,-0.03] | [-0.17,-0.003] | [-0.19,-0.02] |
| Highest | -0.27*** | -0.26*** | -0.24*** | -0.26*** | -0.24*** |
| | [-0.34,-0.19] | [-0.34,-0.18] | [-0.32,-0.17] | [-0.34,-0.18] | [-0.32,-0.16] |
| *Weight parameters* | | | | | |
| Origin | 0.34*** | 0.62*** | 0.62*** | 0.52** | 0.56*** |
| | [0.19,0.51] | [0.33,0.91] | [0.35,0.89] | [0.16,0.87] | [0.24,0.87] |
| Destination | 0.65*** | 0.38** | 0.38** | 0.48** | 0.44** |
| | [0.49,0.81] | [0.09,0.67] | [0.11,0.65] | [0.13,0.84] | [0.13,0.76] |
| *Social mobility* | | | | | |
| Upward | ----- | -0.09* | -0.09* | ----- | ----- |
| | ----- | [-0.18,-0.01] | [-0.18,-0.01] | ----- | ----- |
| Downward | ----- | 0.05 | 0.03 | ----- | ----- |
| | ----- | [-0.05,0.14] | [-0.06,0.12] | ----- | ----- |
| Short-range upward | ----- | ----- | ----- | -0.10* | -0.10* |
| | ----- | ----- | ----- | [-0.19,-0.01] | [-0.19,-0.01] |
| Long-range upward | ----- | ----- | ----- | -0.05 | -0.07 |
| | ----- | ----- | ----- | [-0.17,0.07] | [-0.18,0.04] |
| Short-range downward | ----- | ----- | ----- | 0.05 | 0.04 |
| | ----- | ----- | ----- | [-0.06,0.15] | [-0.06,0.14] |
| Long-range downward | ----- | ----- | ----- | 0.01 | -0.00 |
| | ----- | ----- | ----- | [-0.11,0.13] | [-0.11,0.11] |
| *Sociodemographic controls* | | | | | |
| Age | 0.05*** | 0.05*** | 0.05*** | 0.05*** | 0.05*** |
| | [0.03,0.07] | [0.03,0.07] | [0.03,0.07] | [0.03,0.07] | [0.03,0.07] |
| Male | 0.30*** | 0.30*** | 0.31*** | 0.30*** | 0.31*** |
| | [0.24,0.36] | [0.24,0.36] | [0.25,0.36] | [0.24,0.36] | [0.25,0.37] |
| *Race/ethnicity (ref. White)* | | | | | |
| Black | ----- | ----- | 0.18*** | ----- | 0.18*** |
| | ----- | ----- | [0.11,0.25] | ----- | [0.11,0.25] |
| Hispanic | ----- | ----- | 0.08 | ----- | 0.07 |
| | ----- | ----- | [-0.02,0.17] | ----- | [-0.02,0.17] |
| Other | ----- | ----- | 0.07 | ----- | 0.07 |
| | ----- | ----- | [-0.07,0.21] | ----- | [-0.07,0.21] |
| Married (ref. unmarried) | ----- | ----- | -0.00 | ----- | -0.00 |
| | ----- | ----- | [-0.06,0.06] | ----- | [-0.06,0.05] |
| Rural | ----- | ----- | 0.08* | ----- | 0.08* |
| | ----- | ----- | [0.02,0.14] | ----- | [0.02,0.14] |
| AIC | 13133.67 | 13131.96 | 12723.73 | 13135.00 | 12726.91 |
| BIC | 13191.80 | 13203.01 | 12826.68 | 13218.97 | 12842.73 |

*(Continued)*

**Table 3.** (Continued)

|  | Model 1 | Model 2 | Model 3 | Model 4 | Model 5 |
|---|---|---|---|---|---|
| Observations | 4713 | 4713 | 4713 | 4713 | 4713 |

Notes:

[*] $p < 0.05$,

[**] $p < 0.01$,

[***] $p < 0.001$,

95% confidence intervals in parentheses.

ethnicity, urban/rural divide, and marital status. The results confirm the general patterns observed in Table 2. It is important to clarify that the coefficients for immobile individuals represent weighted mean values of AL for those who occupy diagonal cells in our two-dimensional five by five matrix. In all models, immobile individuals in the highest and lowest socioeconomic quintile, respectively, have significantly lower and higher AL scores than immobile individuals in the middle socioeconomic quintile.

The calculated weight parameters in Model 1 show that the relative importance of parental socioeconomic position (0.35, CI 0.19,0.51) is lower than the importance of individuals' own socioeconomic position (0.65, CI 0.49,0.81), which means that almost twice as much variation in the outcome variable is explained by individuals' destination than by their origin. However, after social mobility variables are introduced, especially when short- and long-range mobility experiences are disentangled in Models 4–5, individuals' own socioeconomic position becomes roughly equal to the effect of parental socioeconomic position. Models also show that age and gender are associated with individuals' AL level. In Model 3, being male and older by one year are both linked to worse health by, respectively, 0.31 (CI 0.25,0.36) and 0.05 (CI 0.03,0.07) standard deviations of AL score.

Model 2 shows that upward mobility is significantly and negatively associated with individuals' AL score, while the variable for downward mobility has a positive sign but it is not statistically significant. These mobility effects are unaffected when further sociodemographic controls—individuals' race/ethnicity, marital status, and urban/rural divide—are accounted for in Model 3. The results for these variables suggest that there are significant differences between Blacks and Whites, the former having more than a 0.18 (CI 0.11,0.25) higher AL score. Marital status does not play a significant role in explaining variation in our outcome measure but living in rural areas is associated with a higher AL score (0.08, CI 0.02,0.14). In Models 4–5, we disentangle social mobility variables into short- and long-range upward and downward mobility. Confidence intervals for three out of four social mobility variables overlap with zero, but a significant association is maintained for short-range upward mobility. Moving up by one socioeconomic quintile is linked with a 0.10 (CI -0.19,-0.01) decrease in individuals' AL score.

In Supporting information, S4 Table, we estimated the effects of upward and downward mobility in reference to both immobility and mobility in the opposite direction by removing the latter coefficients from the fitted models. In S5 Table of supporting information we also present DRM estimates which account for neighbourhood characteristics where individuals lived at Wave I, such as poverty rates, unemployment levels, and race/ethnicity composition. These supplementary results are identical to those reported in the main analysis. Lastly, it is possible that the respondents are still too young to be certain that downward mobility will not change as time passes. This may be particularly true in terms of occupational attainment. To address this issue, in Supporting information, S6 Table, we estimate our models with

education as the only SEP measure and educational mobility parameters. These results show no educational mobility effects, while in terms of health gradient and the importance of the relative weight, no major differences were observed.

### 3.3. Do origin and mobility effects vary by sociodemographic groups?

In Table 4, we estimated DRMs with the interaction terms between the origin weight, social mobility, and a set of covariates—age, gender, race/ethnicity, marital status, and urban-rural divide. We did not find that any interaction terms between the social origin weight and the described parameters were statistically significant, which implied that the effect of origin socioeconomic position on individuals' AL score did not vary by social mobility experiences and sociodemographic characteristics. In Table 4, we also interacted upward and downward mobility experiences with gender, age, marital status, and race/ethnic variables to test if the effect of social mobility on AL score had varying implications for these sociodemographic groups. Past research on various health outcomes in the United States indicates that due to

**Table 4. Origin weights and point estimates for interaction terms from DRM on AL levels.**

|  | Origin weight | Interaction terms | | | | |
|---|---|---|---|---|---|---|
|  |  | Origin weight | *Social mobility* | | | |
|  |  |  | Short-range upward | Long-range upward | Short-range downward | Long-range downward |
| Short-range upward | 0.41* | 1.27 | ---- | ---- | ---- | ---- |
|  | [0.07,0.74] | [-0.19,2.73] | ---- | ---- | ---- | ---- |
| Long-range upward | 0.52* | 0.10 | ---- | ---- | ---- | ---- |
|  | [0.10,0.93] | [-0.59,0.80] | ---- | ---- | ---- | ---- |
| Short-range downward | 0.59** | -0.17 | ---- | ---- | ---- | ---- |
|  | [0.22,0.96] | [-1.07,0.73] | ---- | ---- | ---- | ---- |
| Long-range downward | 0.72*** | -0.51 | ---- | ---- | ---- | ---- |
|  | [0.38,1.05] | [-1.11,0.09] | ---- | ---- | ---- | ---- |
| Age | 1.19 | -0.04 | -0.03 | -0.002 | 0.02 | 0.02 |
|  | [-0.58,2.98] | [-0.16,0.07] | [-0.07,0.02] | [-0.05,0.04] | [-0.02,0.06] | [-0.02,0.06] |
| Male | 0.56** | 0.02 | 0.07 | 0.03 | 0.01 | -0.00 |
|  | [0.20,0.89] | [-0.33,0.38] | [-0.08,0.22] | [-0.12,0.19] | [-0.13,0.15] | [-0.14,0.14] |
| Race/ethnicity |  |  |  |  |  |  |
| Black | 0.48** | 0.33 | 0.05 | 0.03 | -0.00 | -0.08 |
|  | [0.17,0.79] | [-0.06,0.72] | [-0.14,0.24] | [-0.16,0.23] | [-0.17,0.16] | [-0.24,0.08] |
| Hispanic | 0.58* | -0.52 | 0.05 | -0.17 | 0.16 | -0.00 |
|  | [0.27,0.89] | [-1.19,0.14] | [-0.18,0.29] | [-0.39,0.04] | [-0.09,0.42] | [-0.29,0.29] |
| Other | 0.60 | -0.68 | 0.22 | -0.15 | 0.06 | 0.12 |
|  | [0.30,0.92] | [-1.50,0.13] | [-0.58,0.15] | [-0.52,0.22] | [-0.27,0.39] | [-0.22,0.46] |
| Married | 0.42* | 0.26 | -0.00 | 0.17* | -0.10 | 0.06 |
|  | [0.09,0.61] | [-0.09,0.61] | [-0.15,0.15] | [0.01,0.32] | [-0.24,0.04] | [-0.08,0.21] |
| Rural | 0.51** | 0.17 | 0.07 | 0.06 | 0.06 | -0.04 |
|  | [0.19,0.83] | [-.19,0.54] | [-0.10,0.23] | [-0.10,0.23] | [-0.09,0.22] | [-0.20,0.11] |

Notes:

* p < 0.05,

** p < 0.01,

*** p < 0.001.

95% confidence intervals in parentheses, constitutive terms of interactions are not shown, number of observations– 4,713.

historical and structural factors, including discrimination, racial/ethnic differences in social mobility's effect on AL may be present [84,90].

For none of these interaction terms, except marital status, we found statistically significant associations, which indicated that mobility effects on AL did not, as a rule, vary by the considered sociodemographic groups. It should be emphasised that these findings, mostly insignificant, should be interpreted with caution as for the selected interactions the sample size may be too small to provide sufficient variation and conclusive findings.

## 3.4. How does health selection matter?

We now address the possibility that individuals' health before experiencing social mobility affected both mobility trajectories and their AL score. To account for individuals' initial health status in our models, we used their self-rated health (a binary variable which equalled to 1 if respondents' health was worse than very good), BMI levels (with middle quintile of BMI being the reference category), and having chronic health problems at Wave I (a dummy variable). First, we checked if initial health was associated with social mobility in Supporting information, S7 Table, and concluded that those with worse health had, respectively, lower and higher chances to experience upward and downward social mobility, even after the origin socioeconomic position was controlled for. This suggested that health selection has to be accounted for when studying the effects of social mobility on health. Results in Table 5, Models 1 and 2, in turn, show that poor health and high BMI levels prior to social mobility experience were significantly and positively associated with AL score at Wave IV. In comparison to the results in Table 3, we observed a decrease in the size of coefficients for the highest and lowest socioeconomic quintiles of immobile individuals. The earlier detected effect of short-range upward mobility became insignificant in these two models.

Models 3 to 6 show the results where the total sample was split by individuals' self-rated poor and good health at Wave I. This exercise created two groups of individuals, one with initial self-rated excellent/very good (69%) and another with worse than very good (31%) health. For individuals with very good and excellent initial health, AL score of immobile groups did not significantly change in comparison to the pooled sample, but for those with worse initial self-rated health social gradient in AL was less salient. More importantly, we found that short-range upward mobility had a significant negative association with AL score but only among individuals with initial poor health, even after the initial BMI level and chronic health problems were accounted for.

To further explore the role of initial health in the observed positive effect of upward mobility, in Supporting information, S8 Table, we interacted individuals' upward and downward mobility trajectories with their initial health status. Results showed that the interaction term was negative and statistically significant only for those who experienced short-range upward mobility, which again confirmed the positive health implications of upward mobility for the less healthy individuals at Wave I.

## 4. Discussion

In this study, we explored how health outcomes of young adults in the United States are conditioned by individuals' socioeconomic position and their movements between origin and destination, also accounting for initial health and health selection effects into specific mobility experiences. The allostatic load score used in this study is one of the most comprehensive measures of health, especially among relatively young individuals who tend to rate their health as excellent or very good in social surveys, rarely experience hospitalization, and have very low levels of mortality. While social gradient in health is not surprising and there is a number of

**Table 5. Point estimates from DRM on AL levels with initial health.**

| | Model 1 | Model 2 | Model 3—Good health | Model 4—Bad health | Model 5—Good health | Model 6—Bad health |
|---|---|---|---|---|---|---|
| Immobile socioeconomic quintiles | | | | | | |
| Lowest | 0.10** | 0.13*** | 0.12*** | 0.05 | 0.18*** | 0.06 |
| | [0.03,0.16] | [0.06,0.20] | [0.06,0.18] | [-0.07,0.16] | [0.09,0.26] | [-0.07,0.19] |
| Middle-low | 0.04 | 0.05 | 0.09** | 0.03 | 0.05 | 0.08 |
| | [-0.05,0.12] | [-0.03,0.14] | [0.03,0.14] | [-0.12,0.18] | [-0.05,0.15] | [-0.09,0.24] |
| Middle | 0.08* | 0.11** | 0.02 | 0.19* | 0.06 | 0.21** |
| | [0.01,0.15] | [0.03,0.18] | [-0.03,0.06] | [0.03,0.35] | [-0.03,0.14] | [0.05,0.37] |
| Middle-high | -0.04 | -0.09* | -0.09** | 0.05 | -0.10* | -0.06 |
| | [-0.12,0.04] | [-0.17,-0.01] | [-0.16,-0.03] | [-0.15,0.24] | [-0.20,-0.01] | [-0.27,0.15] |
| Highest | -0.18*** | -0.20*** | -0.13*** | -0.32*** | -0.18*** | -0.29** |
| | [-0.25,-0.11] | [-0.28,-0.12] | [-0.19,-0.07] | [-0.48,-0.15] | [-0.26,-0.09] | [-0.49,-0.08] |
| Weight parameters | | | | | | |
| Origin | 0.43* | 0.56*** | -0.30 | 0.37 | 0.60 | 0.51* |
| | [0.05,0.82] | [0.23,0.90] | [-1.30,0.69] | [-0.03,0.78] | [-0.00,1.20] | [0.08,0.93] |
| Destination | 0.57** | 0.44** | 1.30* | 0.63** | 0.40 | 0.49* |
| | [0.18,0.95] | [0.10,0.77] | [0.31,2.30] | [0.22,1.03] | [-0.20,1.00] | [0.07,0.92] |
| Social mobility | | | | | | |
| Short-range upward | -0.08 | -0.08 | 0.01 | -0.20* | -0.02 | -0.26** |
| | [-0.16,0.01] | [-0.17,0.01] | [-0.10,0.12] | [-0.38,-0.03] | [-0.13,0.09] | [-0.44,-0.07] |
| Long-range upward | -0.02 | -0.06 | 0.13 | -0.07 | -0.04 | -0.11 |
| | [-0.11,0.08] | [-0.16,0.05] | [-0.05,0.32] | [-0.23,0.10] | [-0.20,0.11] | [-0.29,0.07] |
| Short-range downward | 0.00 | 0.03 | -0.04 | -0.04 | 0.05 | -0.02 |
| | [-0.08,0.09] | [-0.06,0.13] | [-0.15,0.07] | [-0.19,0.11] | [-0.07,0.17] | [-0.20,0.16] |
| Long-range downward | -0.03 | -0.01 | -0.11 | -0.12 | 0.04 | -0.10 |
| | [-0.12,0.07] | [-0.12,0.09] | [-0.30,0.07] | [-0.26,0.03] | [-0.12,0.20] | [-0.28,0.07] |
| Health selection | | | | | | |
| Initial worse than very good self-rated health | 0.09** | 0.09** | ---- | ---- | ---- | ---- |
| | [0.03,0.15] | [0.03,0.15] | ---- | ---- | ---- | ---- |
| Initial BMI (Ref. = middle quintile) | | | | | | |
| Lowest | -0.33*** | ---- | -0.29*** | -0.44*** | ---- | ---- |
| | [-0.41,-0.25] | ---- | [-0.39,-0.20] | [-0.61,-0.27] | ---- | ---- |
| Low | -0.19*** | ---- | -0.16*** | -0.24** | ---- | ---- |
| | [-0.27,-0.11] | ---- | [-0.25,-0.07] | [-0.42,-0.07] | ---- | ---- |
| High | 0.22*** | ---- | 0.19*** | 0.30*** | ---- | ---- |
| | [0.14,0.30] | ---- | [0.10,0.29] | [0.14,0.46] | ---- | ---- |
| Highest | 0.70*** | ---- | 0.69*** | 0.72*** | ---- | ---- |
| | [0.62,0.78] | ---- | [0.59,0.79] | [0.57,0.86] | ---- | ---- |
| Initial chronic health experience | ---- | 0.13 | ---- | ---- | 0.13 | 0.09 |
| | ---- | [-0.07,0.32] | ---- | ---- | [-0.13,0.38] | [-0.22,0.41] |
| AIC | 11557.95 | 12659.69 | 7825.42 | 3731.87 | 8410.94 | 4224.18 |
| BIC | 11705.35 | 12788.38 | 7958.31 | 3846.92 | 8526.11 | 4324.26 |

(*Continued*)

**Table 5.** (Continued)

| | Model 1 | Model 2 | Model 3—Good health | Model 4—Bad health | Model 5—Good health | Model 6—Bad health |
|---|---|---|---|---|---|---|
| Observations | 4713 | 4713 | 3228 | 1485 | 3228 | 1485 |

Notes:

* p < 0.05,

** p < 0.01,

*** p < 0.001.

95% confidence intervals in parentheses.

studies investigating the role of various aspects of socioeconomic position for AL in the United States (some also using the same dataset as we did [10,60,68,70]), our paper makes an important scholarly contribution as it identifies the relative impact of social origin (parental characteristics) in comparison to the social destination (own attained socioeconomic position); detects possible independent effects of movements from parental to own socioeconomic position; tests if position and mobility effects differ by socioeconomic groups; and investigates the role of health selection in the observed associations.

Using Add Health data and DRM statistical approach, which was specifically designed to distinguish the effects of origin and destination socioeconomic positions from independent effects of mobility between these two, we showed that the combined measure of educational and occupational attainment is a robust predictor of biological markers of health. Those in higher and lower quantiles of socioeconomic position had respectively lower and higher levels of AL even when the standard sociodemographic factors such as age, gender, race/ethnicity, marital status, and rural-urban divide were accounted for. In addition to demonstrating socioeconomic gradient in AL, we revealed that when social mobility is accounted for, socioeconomic origins and destination are of almost equal importance for health. This finding is in line with the results from the United Kingdom where the effects of both parental and own socioeconomic position on individuals' AL were roughly equal [32]. Yet, it contrasts with research on other health and wellbeing outcomes where origin characteristics are less important [80,91]. One explanation for this could be that our AL index based on the selected biomarkers including metabolic and cardiovascular system functioning, is more sensitive to lifetime exposures and experiences, while alternative measures such as, for instance, health-related behaviours and perceptions are more likely to be shaped by individuals' contemporary conditions.

While exploring social mobility effects, we found that upward mobility is linked to lower levels of AL, however, in contrast to theoretical expectations [27,29], downward mobility was not associated with health. These are not completely novel findings, as similar positive effects of upward mobility were shown in the previous studies [20,92]. When we disaggregated mobility experiences in short- and long-range upward mobility, we found that the positive effect on health stemmed from the short-range upward mobility experience. The magnitude of the short-range upward mobility' effect on AL is quite large and it is roughly comparable to the effect of living in urban areas or being two years younger, while for those with the initial poor health this effect is even larger. It is important to note that due to relatively young age composition of the analytical sample, for many individuals the downward social mobility observed in this study may be temporary, which could be one possible explanation why we find a null effect of downward mobility on AL.

Returning to the initial question posed in the introductory section of this study—which theoretical perspectives on the health implications of social mobility do these findings support?

Our results are in line with predictions of "rags to riches" thesis which suggests that psychological benefits of overcoming constraints and moving up in social hierarchy may lead to lower levels of stress and positive life experiences which, in turn, can prevent an increase of AL among individuals. At the same time, our finding that the long-range upward mobility did not reduce AL score could indicate that more extensive mobility experiences may be associated with some negative health-related consequences, as predicted by the dissociation thesis. A more effortful upward mobility experience likely dilutes the positive health implications of moving up in the social hierarchy. This is supported by the observation that when we collapsed socioeconomic positions variables into tertiles rather than quintiles in S9 Table, supporting information, we did not see this positive effect of upward mobility on AL score because mobility between tertiles might simultaneously imply elements of short- and long-range social mobility.

We did not detect heterogeneous effects of social origins and mobility experiences across sociodemographic groups. Apparently, both parental characteristics and mobility from origin to destination have similar implications for health, irrespective of individuals' personal characteristics. Our finding contrast with results from past research using the same dataset, suggesting that selected minority groups (Black and Hispanic individuals) experience higher metabolic syndrome after college completion [49]. Three main methodological aspects are likely to explain this difference. First, our research strategy is focused on disentangling mobility effect from origin and destination effects, based on SEP derived from educational and occupational attainment, and using DRM approach, while Gaydosh et al. rely only on educational mobility and use conventional Poisson regressions. Second, health measures also differ noticeably as Gaydosh et al. rely on metabolic syndrome based on blood pressure, glycosylated hemoglobin, body to waist ratio and cholesterol, while AL measure used in our study in addition to components such as blood pressure and cholesterol includes other biomarkers (i.e. CRP and BMI). Third, Gaydosh et al. use the restricted-full sample, while we use the public version of Add Health. These differences make any direct comparison across the two studies difficult.

Furthermore, we found that accounting for initial health, e.g. health selection, before experiencing social mobility mattered for health outcomes following upward mobility experiences. First, once initial health was controlled, the positive effect of the upward mobility became insignificant. However, when we disaggregated the analytical sample into two groups with good and poor initial health, upward mobility was significantly associated with lower AL score among individuals with poor initial health. One interpretation of this finding is that individuals who had some initial health problems, and presumably high levels of AL, particularly benefited psychologically from moving up in social hierarchy because, regardless of initial adverse conditions, they managed to overcome health-related constraints and consequently developed feelings of life control, achievement, and gratitude. This explanation is also supported by earlier evidence that education is more beneficial for health of those with the lowest propensity to attain higher education based on a wide range of childhood adversities [37].

Despite using a high-quality data set, a comprehensive indicator of socioeconomic position, a validated measure of individuals' health, and DRM—a recommended method to identify the health consequences of social mobility, our study has its limitations. First, due to the age composition of Add Health dataset, we focused on the health implications of social mobility on individuals at a relatively early stage in terms of their life-time socioeconomic position. Although a recent study implies that social mobility might have only short-term consequences observed when mobile individuals are relatively young [39], it is likely that the Add Health cohort at Wave IV is still too young to fully differentiate between mobile and immobile individuals as they are expected to experience further upward or downward social mobility. This seems to be particularly important in terms of occupational attainment and it may be

addressed in the future with Add Health's Wave V data, when respondents are 10 years older. Wave V can also provide valuable health information allowing us re-examining the relationship between social mobility and AL of the relatively older cohort. Methodologically, it is important to note that although we mitigated the bias from confounding factors and health selection, our approach cannot identify a causal relationship between social mobility and AL.

The main conclusion of our study is that not only does socioeconomic position matter for young adults health in the United States, but also whether the attained status is the result of mobility between individuals' origin and destination socioeconomic environments. This mobility effect, however, can be identified if an appropriate statistical approach is employed and individuals' initial health is adequately accounted for. We also call for more nuanced consideration of the impact of short- and long-term mobility on health. The finding that the short-range upward social mobility is associated with better outcomes among those with poor childhood health suggests that social mobility might be a transformative experience for particularly disadvantaged individuals. Yet, our finding that there is no positive health effect of long-range upward mobility also indicates that more extensive mobility experiences might entail costs which cannot be compensated by positive implications of social mobility.

## Supporting information

**S1 Table. Descriptive statistics of AL components.** *Note*: Number of observations—4,713.
(DOCX)

**S2 Table. Correlations between AL levels and Wave I health measures.** *Note*: Number of observations—4,713.
(DOCX)

**S3 Table. Point estimates from OLS models on AL levels.** *Notes*: * $p < 0.05$, ** $p < 0.01$, *** $p < 0.001$, 95% confidence intervals in parentheses.
(DOCX)

**S4 Table. Point estimates from DRM on AL levels, only accounting for upward or downward mobility at a time.** *Notes*: * $p < 0.05$, ** $p < 0.01$, *** $p < 0.001$, 95% confidence intervals in parentheses.
(DOCX)

**S5 Table. Point estimates from DRM on AL levels with neighbourhood controls.** *Notes*: * $p < 0.05$, ** $p < 0.01$, *** $p < 0.001$, 95% confidence intervals in parentheses. Neighbourhood controls capture the most localized available contextual information (approx. 452 housing units, block groups) of the areas in which individuals lived at Wave IV. For modal neighbourhood race categories include white (ref.), black, other; poverty measure represents the proportion of persons with income below poverty level. Based on the distribution of proportion of persons below poverty level in 1989. Low = neighbourhoods where the proportion of the population with income below poverty level was less than 11.6%, the median proportion; medium = between 11.6% and 23.9%; and high (ref.) = above 23.9%; for unemployment rate: Low = neighbourhoods with an unemployment rate less than 6.5%, the median rate, medium = between 6.5% and 10.9%; high = 10.9% or higher.
(DOCX)

**S6 Table. Point estimates from DRM on social mobility.** *Notes*: * $p < 0.05$, ** $p < 0.01$, *** $p < 0.001$, 95% confidence intervals in parentheses.
(DOCX)

**S7 Table. Point estimates from DRM on AL levels with initial health and its interactions with social mobility parameters.** *Notes*: * $p < 0.05$, ** $p < 0.01$, *** $p < 0.001$. 95% confidence intervals in parentheses.
(DOCX)

**S8 Table. Point estimates from DRM, educational attainment only, on AL levels.** *Notes*: * $p < 0.05$, ** $p < 0.01$, *** $p < 0.001$, 95% confidence intervals in parentheses.
(DOCX)

**S9 Table. Point estimates from DRM on AL levels with socioeconomic tertiles and the sample of individuals with poor initial health at Wave I.** *Notes*: * $p < 0.05$, ** $p < 0.01$, *** $p < 0.001$, 95% confidence intervals in parentheses.
(DOCX)

## Author Contributions

**Conceptualization:** Alexi Gugushvili, Grzegorz Bulczak, Olga Zelinska, Jonathan Koltai.

**Data curation:** Alexi Gugushvili, Grzegorz Bulczak.

**Formal analysis:** Alexi Gugushvili, Grzegorz Bulczak, Olga Zelinska, Jonathan Koltai.

**Funding acquisition:** Alexi Gugushvili.

**Investigation:** Alexi Gugushvili, Grzegorz Bulczak, Olga Zelinska, Jonathan Koltai.

**Methodology:** Alexi Gugushvili, Grzegorz Bulczak.

**Project administration:** Alexi Gugushvili, Olga Zelinska.

**Software:** Alexi Gugushvili, Grzegorz Bulczak.

**Supervision:** Alexi Gugushvili.

**Writing – original draft:** Alexi Gugushvili, Grzegorz Bulczak, Olga Zelinska, Jonathan Koltai.

**Writing – review & editing:** Alexi Gugushvili, Olga Zelinska, Jonathan Koltai.

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
