## [Decision Letter · Decision Letter 0]

31 Mar 2021

PONE-D-21-03406

Socioeconomic Position, Social Mobility, and Health Selection Effects on Allostatic Load In the United States

PLOS ONE

Dear Dr. Gugushvili,

Thank you for submitting your manuscript to PLOS ONE. After careful consideration, we feel that it has merit but does not fully meet PLOS ONE’s publication criteria as it currently stands. Therefore, we invite you to submit a revised version of the manuscript that addresses the points raised during the review process.

I have read your manuscript and received input from three expert reviewers. All reviewers note key strengths of the study but also raise important concerns. Reviewers 2 and 3 both note how findings reported here on race differences in the associations between social mobility and allostatic load differ from previously published findings in the same cohort. In the revision, please include discussion of this discrepancy and likewise respond to comments from Reviewer 2 regarding statistical power to test interactions by race and how Hispanic ethnicity was treated. Each reviewer requested additional methodological details, and the authors are especially encouraged to provide more information regarding the operationalization of social mobility, conceptualization of allostatic load, and the interpretation of coefficients from diagonal reference models. Finally, please ensure that the study is reported in accordance with guidelines from STROBE (https://www.strobe-statement.org/), in particular that the study design is included in the title or abstract and that limitations of observational research are acknowledged, including in the Abstract.

We look forward to receiving your revised manuscript.

Kind regards,

Jennifer Morozink Boylan

Academic Editor

PLOS ONE

Journal Requirements:

Reviewers' comments:

Reviewer's Responses to Questions

**Comments to the Author**

1. Is the manuscript technically sound, and do the data support the conclusions?

Reviewer #1: Partly

Reviewer #2: Yes

Reviewer #3: Yes

2. Has the statistical analysis been performed appropriately and rigorously? 

Reviewer #1: N/A

Reviewer #2: Yes

Reviewer #3: I Don't Know

3. Have the authors made all data underlying the findings in their manuscript fully available?

Reviewer #1: Yes

Reviewer #2: Yes

Reviewer #3: Yes

4. Is the manuscript presented in an intelligible fashion and written in standard English?

Reviewer #1: Yes

Reviewer #2: Yes

Reviewer #3: Yes

5. Review Comments to the Author

Reviewer #1: This paper examined the effect of social mobility on AL using the data from Add Health Study. The authors found that short-range upward mobility had a positive impact on health, and such a health benefit gain associated with social upward mobility was only observed for participants reporting poor health during adolescence. Overall, the strength of this paper is the inclusion of a large, national sample. However, there were also a couple issues of this current form. My major concern is that the lack of AL data at baseline may not able to completely differentiate between health selection and social causation effects in this study, thought the authors included self-reported health in the model. However, self-reported health may not always reflect the objective health status. The variances of AL that was explained by self-reported health was relatively low. Another concern is that participants were relatively young and that their social standings relative to their patients may more likely to be subjective to change. That is the downward social mobility observed in this study may likely to be temporary, and that is could be one possible explanation that this study observed a null effect of downward mobility on AL. Other issues were detailed below.

1. A recent article examined the effects of social mobility on inflammation may be helpful to discuss the introduction of social mobility and health.

Surachman, A., Rice, C., Bray, B., Gruenewald, T., & Almeida, D. (2020). Association between socioeconomic status mobility and inflammation markers among White and black adults in the United States: a latent class analysis. Psychosomatic medicine, 82(2), 224-233.

2. It is unclear how short-range and long-range were operationalized. To what extent of the changes in SES was referred to as one-step mobility?

3. It is unclear how self-reported heath was assessed. Was it assessed using one or multiple items? What were the response options?

4. Medical chronic health conditions may affect both educational attainment and occupation, as well as AL. It would be interesting to know if results reported to this study would be robust after controlling for medical chronic health conditions.

5. In Table 2, please specify how low and high SES quintiles were operationalized.

6. The author stated in the limitation section that the upward or downward social mobility observed in this study may change and this is may be particular for occupational attainment. I would like to know if the results observed this study would be similar when SES is assessed using educational attainment as the single indictor. This may help to shed some light on this issue.

Reviewer #2: This manuscript investigates the influence of childhood and adult socioeconomic status on health using public data from Add Health and diagonal reference models. The authors find that short range upward mobility is associated with improved health, while long range upward mobility and downward mobility are not significantly associated with health. They find no evidence of heterogeneity by sociodemographic characteristics.

I have two major comments. First, the authors construct a measure of health from biomarker data. They call this measure allostatic load. Allostatic load, as the authors state as well (203), is a measure of neuroendocrine, immune, metabolic and cardiovascular function. The authors only use measures of metabolic and cardiovascular function (as well as one inflammatory measure). They would be better served calling this a measure of cardiometabolic health, rather than allostatic load. More generally, there is a need to more fully introduce and motivate the measure of health that is used and why it is appropriate for the given analysis. The authors also mention that self-rated health is limited and allostatic load addresses these limitations, but then rely on self-rated health to account for health selection. Why not use a measure of BMI or childhood illnesses to test for health selection? Or temper the discussion of self-rated health. Moreover, in their discussion the authors state (442): “The main explanation for this could be that our AL index based on neuroendocrine, immune, metabolic, and cardiovascular system functioning, is more sensitive to lifetime exposures and experiences, while alternative measures such as, for instance, health-related behaviours and perceptions are more likely to be shaped by individuals’ contemporary conditions.” This is an incorrect description of the measure of AL that they were able to construct.

Second, the authors conduct tests of moderation in the relationship between SES and health by race. This analysis is important, but the authors do not appropriately motivate the analysis. There is a large literature that is not cited, including foundational work by Sherman James on John Henryism, and more recent work on skin-deep resilience by Brody, Miller and Chen. Moreover, the racial categories constructed are unusual for the US, where Hispanic ethnicity is usually accounted for in some way. Are the Black and White groups including Hispanic ethnicity here? I’d suggest the authors also read this recent article on how to discuss results on racial disparities in health: https://www.healthaffairs.org/do/10.1377/hblog20200630.939347/full/?utm_medium=social&utm_sour=& . Note the language used on page 452 – talking about the health effects of race. I also wonder whether, given the use of the public data in Add Health and the limited sample size, if the authors are adequately powered to estimate these interactions. The findings are much less conclusive if this is the case.

A few minor points:

- Wave V of Add Health is now available. Why not use? This would alleviate concern over early adult SES.

- Line 176 – testing if position and mobility effects differ by socioeconomic groups – what are the socioeconomic groups? Do they mean sociodemographic characteristics of race, gender, etc?

- Settlement type is a strange way to refer to urban/rural in a US context.

- Line 457 – Mentions psychological benefits, but no tests of this in the analysis – goes beyond the data in discussion

- Some references from Add Health that are relevant but not mentioned:

o Brody, Gene H.; Yu, Tianyi; Miller, Gregory E.; & Chen, Edith (2016). Resilience in adolescence, health, and psychosocial outcomes. Pediatrics, 138(6), e20161042. PMCID: PMC5127063

o Chen, Edith; Yu, Tianyi; Siliezar, Rebekah; Drage, Jane N.; Dezil, Johanna; Miller, Gregory E.; & Brody, Gene H. (2020). Evidence for skin-deep resilience using a co-twin control design: Effects on low-grade inflammation in a longitudinal study of youth. Brain, Behavior, and Immunity. PMCID: PMC7415558

o Miller, Gregory E.; Chen, Edith; Yu, Tianyi; & Brody, Gene H. (2020). Youth Who Achieve Upward Socioeconomic Mobility Display Lower Psychological Distress But Higher Metabolic Syndrome Rates as Adults: Prospective Evidence From Add Health and MIDUS. Journal of the American Heart Association, 9(9). PMCID: PMC7428555

o Gaydosh, Lauren; Schorpp, Kristen M.; Chen, Edith; Miller, Gregory E.; & Harris, Kathleen Mullan (2018). College completion predicts lower depression but higher metabolic syndrome among disadvantaged minorities in young adulthood. Proceedings of the National Academy of Sciences, 115(1), 109-114. PMCID: PMC5776811

o Belsky, Daniel W.; Domingue, Benjamin W.; Wedow, Robbee; Arseneault, Louise; Boardman, Jason D.; Caspi, Avshalom; Conley, Dalton; Fletcher, Jason M.; Freese, Jeremy; & Herd, Pamela, et al. (2018). Genetic analysis of social-class mobility in five longitudinal studies. Proceedings of the National Academy of Sciences.

o Yang, Yang Claire; Gerken, Karen; Schorpp, Kristen M.; Boen, Courtney; & Harris, Kathleen Mullan (2017). Early-Life Socioeconomic Status and Adult Physiological Functioning: A Life Course Examination of Biosocial Mechanisms. Biodemography and Social Biology, 63(2), 87-103. PMCID: PMC5439296

Reviewer #3: Manuscript PONE-D-21-03406 “Socioeconomic position, social mobility, and health selection effects on allostatic load in the United States” explores associations between socioeconomic conditions, social mobility, and indicators of physical health (i.e. allostatic load) in a large an well-studied sample of US adults, the Add Health sample, who have been followed from adolescence to young adulthood. Authors observed a negative association between socioeconomic position and allostatic load. They also observed that “origin” and “destination” socioeconomic circumstances were similarly associated with allostatic load and found an association between short-range upward social mobility and lower allostatic load, particularly among individuals who self-reported worse health at the initial wave of Add Health. The manuscript is well written, comprehensive, and the findings are compelling. The findings will be of interest to those concerned with associations between socioeconomic circumstances and health. I only have a few comments.

1. I am not an expert in diagonal reference models (DRMs), so most of my comments concern this method. My feeling after reading the introduction and methods is that much was said about DRMs (pg 4, pg 12-13), but that still more could be said (perhaps even in simpler and more applied terms) to make the unfamiliar reader better understand the rationale for (and interpretation of) DRMs to address questions about social mobility. One applied example provided by the authors is: “These associations could be, at least partially, explained by the fact that the upwardly mobile groups do not include those who ended up in the bottom quintile, while downwardly mobile groups do not include those who ended up in the highest quintile.” (pg 14), but I found this logic confusing because it seems that, by definition, upwardly mobile groups cannot end up in the bottom quintile. So, the less familiar reader is still not clear on what exactly DRMs are and how they can address limitations in the prior literature.

2. For readers less familiar with DRM methods, it would be helpful if there was clearer interpretation about what exactly each type of parameter is indicating in Table 3. For example, in the full model (Model 5), what is the exact interpretation of the coefficient for the intercept? The interpretation of the estimate for “short-range upward” is straightforward enough, given the earlier explanation that “Social mobility variables and associated point estimates,…, in DRM approach could be interpreted in the same way as in a conventional regression model, a reference category being a group of individuals with the same socioeconomic position as their parents.” (pg 13). In Model 5 of Table 3, the point estimate for “Lowest” is 0.17. Is its p-value below 0.001 indicating that this AL estimate of 0.17 is significantly greater than 0 (the mean of AL in the sample)? Similarly, when interpreting the weight parameters in Model 1, authors state that “The calculated weight parameters in Model 1 shows that the relative importance of parental socioeconomic position (0.35, CI 0.19,0.51) is lower than the importance of individuals’ own socioeconomic position (0.65, CI 0.49,0.81).” (pg 14). What is the exact interpretation of a weight of 0.65?

3. In Models 2 and 3 of Table 3, with the addition of the social mobility variables, the “importance” of own socioeconomic position seems to drop far below the importance of parental socioeconomic position. This seems counterintuitive given that there is, simultaneously, a statistically significant association between upward social mobility and AL. Similarly, in Models 4 and 5 of Table 3, Origin and destination weights seem approximately similar, but still, destination “importance” seems smaller than origin. Given that a significant association was found for short-range upward mobility, I would have guessed that destination would have had a slightly stronger weight. Some interpretation of these specific findings would be helpful.

4. In Table 4, authors observed no interaction between race and mobility on AL. This finding would seem to contrast with the findings of Gaydosh et al., 2017, “College completion predicts lower depression but higher metabolic syndrome among disadvantaged minorities in young adulthood”, Proceedings of The National Academy of Sciences, 115(1), 109-114, who, also utilizing Add Health data, observed some physical health costs associated with upward social mobility among Black and Hispanic adults. Of course, the methods utilized in the two studies are different, but interpretation of findings from the current study considering these previous findings in Add Health are needed.

6. PLOS authors have the option to publish the peer review history of their article (what does this mean?). If published, this will include your full peer review and any attached files.

Reviewer #1: No

Reviewer #2: No

Reviewer #3: No

---

## [Author Response · Author response to Decision Letter 0]

18 May 2021

Dear Editor and Reviewers,

Thank you very much for your reviews and giving us the opportunity to revise our manuscript.

We have now completed all suggested and required revisions. We believe that the review process has improved the quality of our study. Below we provide the detailed explanations how we have dealt with all points raised by the reviewers. 

Thank you for your attention on this matter.

Best wishes

Authors of the manuscript (PONE-D-21-03406) 

EDITOR

I have read your manuscript and received input from three expert reviewers. All reviewers note key strengths of the study but also raise important concerns. 

### Thank you very much for your work on our submission. 

Reviewers 2 and 3 both note how findings reported here on race differences in the associations between social mobility and allostatic load differ from previously published findings in the same cohort. In the revision, please include discussion of this discrepancy and likewise respond to comments from Reviewer 2 regarding statistical power to test interactions by race and how Hispanic ethnicity was treated. 

### We now explicitly address the discrepancy between our findings and those reported by Gaydosh et al. Please find our detailed response and explanation after the last comment of Reviewer 3 and see as well the changes in the main text. To briefly summarise here, there are three main methodological reasons (1. different models, 2. different health outcomes, and 3. different samples) which are likely to explain the observed difference. As for statistical power to test interactions, we now explicitly state that our findings should be interpreted with caution as for the selected interactions the sample size may be too small to provide sufficient variation and conclusive findings. 

Each reviewer requested additional methodological details, and the authors are especially encouraged to provide more information regarding the operationalization of social mobility, conceptualization of allostatic load, and the interpretation of coefficients from diagonal reference models. 

### As you can see below in our answers to reviewers’ comments, we now provide more information in Research Design section on operationalization of social mobility, conceptualization of allostatic load, and the interpretation of coefficients from diagonal reference models. 

Finally, please ensure that the study is reported in accordance with guidelines from STROBE (https://www.strobe-statement.org/), in particular that the study design is included in the title or abstract and that limitations of observational research are acknowledged, including in the Abstract.

### Now abstract clearly states that we use data from the National Longitudinal Study of Adolescent Health’s Waves I and IV and that we are able only to mitigate (not eliminate) health selection concerns in observational data we use.

 

REVIEWER #1: 

This paper examined the effect of social mobility on AL using the data from Add Health Study. The authors found that short-range upward mobility had a positive impact on health, and such a health benefit gain associated with social upward mobility was only observed for participants reporting poor health during adolescence. Overall, the strength of this paper is the inclusion of a large, national sample. However, there were also a couple issues of this current form. 

### Thank you for this assessment. 

My major concern is that the lack of AL data at baseline may not able to completely differentiate between health selection and social causation effects in this study, thought the authors included self-reported health in the model. However, self-reported health may not always reflect the objective health status. The variances of AL that was explained by self-reported health was relatively low. 

### Thank you for pointing this out. The survey does not include AL in Wave I, yet we address your concern on accounting the baseline health by additionally controlling for BMI and chronic health problems (including heart problems, asthma, diabetes and difficulties using limbs) at Wave I. The results are shown now in the updated Table 5 on page 21. Accounting for these additional variables for initial health did not affect our main results. 

Another concern is that participants were relatively young and that their social standings relative to their patients may more likely to be subjective to change. That is the downward social mobility observed in this study may likely to be temporary, and that is could be one possible explanation that this study observed a null effect of downward mobility on AL. Other issues were detailed below.

### Thank you for this note. We completely agree that downward mobility experiences might be followed by upward mobility experiences and this is can be an interesting topic to explore in another study. However, we also argue that health and wellbeing outcomes relating to social mobility among individuals being in their late 20s are a socially important topic to investigate. Still, based on your suggestion, we now include the following note on page 23 in the discussion section:

“It is important to note that due to relatively young age composition of the analytical sample, for many individuals the downward social mobility observed in this study may be temporary, which could be one possible explanation why we find a null effect of downward mobility on AL." 

The issue of downward mobility being only temporary is further examined by studding health outcomes for educational mobility separately in supplementary materials (Table S6). In the case of educational mobility, it can be plausibly assumed that most of the respondents achieved their lifetime attainment. We do not find that downward or upward educational mobility is associated with worse or better health outcomes. 

1. A recent article examined the effects of social mobility on inflammation may be helpful to discuss the introduction of social mobility and health.

Surachman, A., Rice, C., Bray, B., Gruenewald, T., & Almeida, D. (2020). Association between socioeconomic status mobility and inflammation markers among White and black adults in the United States: a latent class analysis. Psychosomatic medicine, 82(2), 224-233.

### Thank you for this. We now cite this reference in the paper, see lines 285-288 on page 12. 

2. It is unclear how short-range and long-range were operationalized. To what extent of the changes in SES was referred to as one-step mobility?

### We explain this now in more details, please see the text in the third paragraph on page 11: 

“Finally, to derive the index of socioeconomic position for parents and individuals, we combined educational and occupational attainment variables. This resulted in scores ranging from 2 to 10 points for the highest achieving individuals. To ensure that each mobility group had adequate representation, we collapsed the combined socioeconomic position scores into quintiles, where quintile 5 represents the top 20% (highest attainment based on educational and occupational status). From these combined measures we calculated intergenerational social mobility variables. We subtracted parental from individuals scores. This resulted in a mobility measure ranging from -4 to 4, where 0 represents the immobile group. For example, if the respondent achieved a score equal 5, highest attainment (top quintile) and parental attainment was equal to 4, the difference between the two scores produces one-step upward mobility. 

To ensure sufficient variation we collapse two, three and four steps into a long-range mobility indicator, separately for upward and downward mobility, while one-step mobility represents short-range mobility.”

3. It is unclear how self-reported heath was assessed. Was it assessed using one or multiple items? What were the response options?

### We now provide a specific wording of the question and answers on self-rated health question on page 12, lines 291-293: 

“More specifically, the respondents were asked the following question: “In general, how is your health?” which they could rate on from 1 (= poor) to 5 (= excellent) health.” 

4. Medical chronic health conditions may affect both educational attainment and occupation, as well as AL. It would be interesting to know if results reported to this study would be robust after controlling for medical chronic health conditions.

### Table 5 in the main results presents findings when Wave 1 chronic health as well as BMI scores are accounted for. This new initial health variables do not significantly change the main results. 

5. In Table 2, please specify how low and high SES quintiles were operationalized.

### We now describe how SEP quintiles were operationalised in greater detail in the main text of the paper on page 11, paragraph 4. More specifically we write that: 

“…to derive the index of socioeconomic position for parents and individuals, we combined educational and occupational attainment variables. This resulted in scores ranging from 2 to 10 points for the highest achieving individuals. To ensure that each mobility group had adequate representation, we collapsed the combined socioeconomic position scores into quintiles, where quintile 5 represents the top 20% (highest attainment based on educational and occupational status). From these combined measures we calculated intergenerational social mobility variables.”

Lines 350-16. The author stated in the limitation section that the upward or downward social mobility observed in this study may change and this is may be particular for occupational attainment. I would like to know if the results observed this study would be similar when SES is assessed using educational attainment as the single indictor. This may help to shed some light on this issue.

### Based on your suggestion, we now introduce additional analysis in supplementary materials in which we only account for mobility in educational attainment. The following text appears on page 21, lines 458-462: 

“It is possible that the respondents are still too young to be certain that downward mobility will not change as time passes. This may be particularly true in terms of occupational attainment. To address this issue, in supplementary materials, Table S6, we estimate our models with education as the only SEP measure and educational mobility parameters. These results show no educational mobility effects, while in terms of health gradient and the importance of the relative weight, no major differences were observed.” 

REVIEWER #2: 

This manuscript investigates the influence of childhood and adult socioeconomic status on health using public data from Add Health and diagonal reference models. The authors find that short range upward mobility is associated with improved health, while long range upward mobility and downward mobility are not significantly associated with health. They find no evidence of heterogeneity by sociodemographic characteristics.

### Thank you. Yes, indeed this largely correct assessment of our study. 

I have two major comments. First, the authors construct a measure of health from biomarker data. They call this measure allostatic load. Allostatic load, as the authors state as well (203), is a measure of neuroendocrine, immune, metabolic and cardiovascular function. The authors only use measures of metabolic and cardiovascular function (as well as one inflammatory measure). They would be better served calling this a measure of cardiometabolic health, rather than allostatic load. 

### Thank you for this comment. We changed the description of AL measure and added additional motivation lines. On page 9, we now first state that AL index may incorporate 

“… neuroendocrine, immune, metabolic, and cardiovascular system functioning and is a validated predictor of morbidity and mortality outcomes, especially at the earlier stages of life.” 

In other words, this sentence implies that AL does not have to incorporate all listed areas of bodily functions. Following this clarification on the same page, lines 220-223, we state that: 

“Our approach to constructing this measure is closely matched with the previous research in which AL is based on lipid and glucose metabolism, inflammation (C-reactive protein and fibrinogen), body fat deposition (body mass index and waist measurement) and cardiovascular measures [30].” 

However, we are happy to substitute the name of our outcome variable to ‘cardiometabolic function’ if the editor thinks we should do so. 

More generally, there is a need to more fully introduce and motivate the measure of health that is used and why it is appropriate for the given analysis. The authors also mention that self-rated health is limited and allostatic load addresses these limitations, but then rely on self-rated health to account for health selection. Why not use a measure of BMI or childhood illnesses to test for health selection? Or temper the discussion of self-rated health. Moreover, in their discussion the authors state (442): “The main explanation for this could be that our AL index based on neuroendocrine, immune, metabolic, and cardiovascular system functioning, is more sensitive to lifetime exposures and experiences, while alternative measures such as, for instance, health-related behaviours and perceptions are more likely to be shaped by individuals’ contemporary conditions.” This is an incorrect description of the measure of AL that they were able to construct.

### Thank you for this comment. We have now changed the description of AL measure and added additional motivation in the first and the first paragraph on page 10. We note that:

“…we consider this measure to be particularly appropriate for our study as it is able to capture even relatively small changes in young adults’ health. This is especially important in the context of social mobility where the sensitivity of health outcome measures has yielded in mixed results.” 

As for using alternative measures of health selection, we note that self-rated health has stronger association with AL than other indicators. This measure also gives us an opportunity to divide the sample into those reporting better and worse health. Nonetheless, based on your suggestion, in the main analysis, Table 5, we now present results with Wave 1 BMI and chronic health problems are accounted for. 

Second, the authors conduct tests of moderation in the relationship between SES and health by race. This analysis is important, but the authors do not appropriately motivate the analysis. There is a large literature that is not cited, including foundational work by Sherman James on John Henryism, and more recent work on skin-deep resilience by Brody, Miller and Chen.

### We have now refer and cite the suggested sources on, among other areas, structural racism, the disproportionate stressors experienced by racialized groups. Please see the second paragraph on page 7, and we also state in the main text on page 19: 

“Past research on various health outcomes in the United States indicates that due to historical and structural factors, including discrimination, racial/ethnic differences in social mobility’s effect on AL may be present [81,87].”

Moreover, the racial categories constructed are unusual for the US, where Hispanic ethnicity is usually accounted for in some way. Are the Black and White groups including Hispanic ethnicity here? I’d suggest the authors also read this recent article on how to discuss results on racial disparities in health: https://www.healthaffairs.org/do/10.1377/hblog20200630.939347/full/?utm_medium=social&utm_sour=& . Note the language used on page 452 – talking about the health effects of race. 

### Thank you for this note and suggestion. We now account for Hispanic ethnicity in the updated race/ethnicity variable which includes the following four categories: Whites (60% of the analytical sample), Blacks (24%), Hispanics (11%) and other category (5%). This operationalisation is similar what Gaydosh et al. use in their study (College completion predicts lower 726 depression but higher metabolic syndrome among disadvantaged minorities in young adulthood. 727 Proc Natl Acad Sci U S A. 2018;115: 109–114). All results are now updated using this new race/ethnicity variable but this does not affect our original findings. 

I also wonder whether, given the use of the public data in Add Health and the limited sample size, if the authors are adequately powered to estimate these interactions. The findings are much less conclusive if this is the case.

### On pages 19-20, lines 435-437, as suggested, we added a point about caution in interpreting these interactions. We state that…

“It should be emphasised that these findings, mostly insignificant, should be interpreted with caution as for the selected interactions the sample size may be too small to provide sufficient variation and conclusive findings.” 

A few minor points:

- Wave V of Add Health is now available. Why not use? This would alleviate concern over early adult SES.

### We started working on this paper quite some time before public version of Wave V became available. Adjusting our analytical sample and the age composition would likely change the framing and results of the study, yet in the discussion section we explicitly state that data from Wave V would be very helpful to understand the effects of social mobility on later health outcomes among Add Health participants. 

- Line 176 – testing if position and mobility effects differ by socioeconomic groups – what are the socioeconomic groups? Do they mean sociodemographic characteristics of race, gender, etc?

### Yes, indeed, we meant to say socioeconomic groups. The following edited text now appears in the end of the introduction section, page 8, lines 181-182: 

“…testing if position and mobility effects differ by sociodemographic characteristics such as gender and race;”

- Settlement type is a strange way to refer to urban/rural in a US context.

### We have now replaced “settlement type” with urban/rural divide throughout the text.

- Line 457 – Mentions psychological benefits, but no tests of this in the analysis – goes beyond the data in discussion

### Thank you for pointing this out. Yes, we are mentioning the possible psychological benefits of upward mobility, but we do this primarily to describe potential mechanisms linking social mobility and health. In this study, we did not intend to test physiological effects of mobility as there are other studies on this topic. 

- Some references from Add Health that are relevant but not mentioned:

o Brody, Gene H.; Yu, Tianyi; Miller, Gregory E.; & Chen, Edith (2016). Resilience in adolescence, health, and psychosocial outcomes. Pediatrics, 138(6), e20161042. PMCID: PMC5127063

o Chen, Edith; Yu, Tianyi; Siliezar, Rebekah; Drage, Jane N.; Dezil, Johanna; Miller, Gregory E.; & Brody, Gene H. (2020). Evidence for skin-deep resilience using a co-twin control design: Effects on low-grade inflammation in a longitudinal study of youth. Brain, Behavior, and Immunity. PMCID: PMC7415558

o Miller, Gregory E.; Chen, Edith; Yu, Tianyi; & Brody, Gene H. (2020). Youth Who Achieve Upward Socioeconomic Mobility Display Lower Psychological Distress But Higher Metabolic Syndrome Rates as Adults: Prospective Evidence From Add Health and MIDUS. Journal of the American Heart Association, 9(9). PMCID: PMC7428555

o Gaydosh, Lauren; Schorpp, Kristen M.; Chen, Edith; Miller, Gregory E.; & Harris, Kathleen Mullan (2018). College completion predicts lower depression but higher metabolic syndrome among disadvantaged minorities in young adulthood. Proceedings of the National Academy of Sciences, 115(1), 109-114. PMCID: PMC5776811

o Belsky, Daniel W.; Domingue, Benjamin W.; Wedow, Robbee; Arseneault, Louise; Boardman, Jason D.; Caspi, Avshalom; Conley, Dalton; Fletcher, Jason M.; Freese, Jeremy; & Herd, Pamela, et al. (2018). Genetic analysis of social-class mobility in five longitudinal studies. Proceedings of the National Academy of Sciences.

o Yang, Yang Claire; Gerken, Karen; Schorpp, Kristen M.; Boen, Courtney; & Harris, Kathleen Mullan (2017). Early-Life Socioeconomic Status and Adult Physiological Functioning: A Life Course Examination of Biosocial Mechanisms. Biodemography and Social Biology, 63(2), 87-103. PMCID: PMC5439296

### At the start of “2.2. Measures” section, we now include the suggested references by noting that: 

“...Numerous past studies used the Add Health to investigate the impact of socioeconomic position on physical [60–62] and mental [47,53,63] health outcomes.” 

REVIEWER #3: 

Manuscript PONE-D-21-03406 “Socioeconomic position, social mobility, and health selection effects on allostatic load in the United States” explores associations between socioeconomic conditions, social mobility, and indicators of physical health (i.e. allostatic load) in a large an well-studied sample of US adults, the Add Health sample, who have been followed from adolescence to young adulthood. Authors observed a negative association between socioeconomic position and allostatic load. They also observed that “origin” and “destination” socioeconomic circumstances were similarly associated with allostatic load and found an association between short-range upward social mobility and lower allostatic load, particularly among individuals who self-reported worse health at the initial wave of Add Health. The manuscript is well written, comprehensive, and the findings are compelling. The findings will be of interest to those concerned with associations between socioeconomic circumstances and health. I only have a few comments.

### Thank you very much for this kind assessment.

1. I am not an expert in diagonal reference models (DRMs), so most of my comments concern this method. My feeling after reading the introduction and methods is that much was said about DRMs (pg 4, pg 12-13), but that still more could be said (perhaps even in simpler and more applied terms) to make the unfamiliar reader better understand the rationale for (and interpretation of) DRMs to address questions about social mobility. One applied example provided by the authors is: “These associations could be, at least partially, explained by the fact that the upwardly mobile groups do not include those who ended up in the bottom quintile, while downwardly mobile groups do not include those who ended up in the highest quintile.” (pg 14), but I found this logic confusing because it seems that, by definition, upwardly mobile groups cannot end up in the bottom quintile. So, the less familiar reader is still not clear on what exactly DRMs are and how they can address limitations in the prior literature.

### Thank you for this observation. Because we did not intend this paper to be methodologically oriented, we do not provide a comprehensive description of DRM approach. However, it is largely accepted in the literature on the consequences of social mobility that the DRM approach is superior to alternatives and we refer readers to some of the recent studies on this topic. Nonetheless, based on your suggestion, we introduce further details in the description of DRM by emphasizing on pages 13-14 that: 

(1) “conventional statistical models cannot simultaneously include origin, destination, and mobility parameters;” 

(2) “off-diagonal cells in two-dimensional table represent specific mobility trajectories;” and 

(3) “an extensive overview of this statistical method, its usefulness in modelling of social mobility effects, and a comparison with conventional regression approaches are described and demonstrated elsewhere”. 

We also add to this section with the following message at the bottom of page 15:

“For various empirical applications of DRM approach in different countries and contexts readers can refer to studies on, among other areas, redistribution preferences [84], likelihood of smoking [85], attitudes toward immigrants [86]. 

As for the statement given on page 14 of the original submission (“These associations could be, at least partially, explained by the fact that the upwardly mobile groups do not include those who ended up in the bottom quintile, while downwardly mobile groups do not include those who ended up in the highest quintile”), we modified it and added further information in the following sentence, lines 362-365, on page 15: 

“The latter also suggests that upwardly and downwardly mobile individuals differ by their social origin and destination positions and to disentangle these position effects from social mobility effects, we employed the above-described statistical approach – DRM.”

2. For readers less familiar with DRM methods, it would be helpful if there was clearer interpretation about what exactly each type of parameter is indicating in Table 3. For example, in the full model (Model 5), what is the exact interpretation of the coefficient for the intercept? 

### Thank you for this clarifying question. As we mentioned earlier in the text, the parameters of DRM models should be interpreted as ordinary regression models, which means that the reported intercept show mean AL for individuals with all covariates being equal to zero. However, considering that the reported coefficients for immobile individuals in respective socioeconomic positions are essentially class-specific intercepts, we decided to remove the global intercept from the models as it does not help in understanding of the models, does not have a substantive meaning, and some of the most relevant studies using DRM approach also do not report global intercepts. See for instance the following articles:

Houle, Jason N. 2011. “The Psychological Impact of Intragenerational Social Class Mobility.” Social Science Research 40(3):757–72.

van der Waal, Jeroen, Stijn Daenekindt, and Willem de Koster. 2017. “Statistical Challenges in Modelling the Health Consequences of Social Mobility: The Need for Diagonal Reference Models.” International Journal of Public Health 62(9):1029–37.

The interpretation of the estimate for “short-range upward” is straightforward enough, given the earlier explanation that “Social mobility variables and associated point estimates,…, in DRM approach could be interpreted in the same way as in a conventional regression model, a reference category being a group of individuals with the same socioeconomic position as their parents.” (pg 13). In Model 5 of Table 3, the point estimate for “Lowest” is 0.17. Is its p-value below 0.001 indicating that this AL estimate of 0.17 is significantly greater than 0 (the mean of AL in the sample)?

### Thank you for this clarifying question. Class coefficients indicate the class-specific intercepts. In other words, the coefficients represent weighted mean values of AL for those who occupy diagonal cells in our two-dimensional five by five matrix. We added more clarifying information on this matter in the first paragraph of Section 3.2 on page 16. 

Similarly, when interpreting the weight parameters in Model 1, authors state that “The calculated weight parameters in Model 1 shows that the relative importance of parental socioeconomic position (0.35, CI 0.19,0.51) is lower than the importance of individuals’ own socioeconomic position (0.65, CI 0.49,0.81).” (pg 14). What is the exact interpretation of a weight of 0.65? 

### Thank you for this comment. Weight greater than 0.5 indicates greater importance of destination characteristics (position) in determining AL. The destination weight (w) ranges between 0 and 1, with 0 indicating that the destination class plays no role for determining current AL and 1 indicating that it is only the destination that explains the variation in AL. The origin weight equals to 1-w. Now, after editing the corresponding section, you can find the following text on page 16, the second paragraph: 

“The calculated weight parameters in Model 1 show that the relative importance of parental socioeconomic position (0.35, CI 0.19,0.51) is lower than the importance of individuals’ own socioeconomic position (0.65, CI 0.49,0.81), which means that almost twice as much variation in the outcome variable is explained by individuals’ destination than by their origin.”

3. In Models 2 and 3 of Table 3, with the addition of the social mobility variables, the “importance” of own socioeconomic position seems to drop far below the importance of parental socioeconomic position. This seems counterintuitive given that there is, simultaneously, a statistically significant association between upward social mobility and AL. Similarly, in Models 4 and 5 of Table 3, Origin and destination weights seem approximately similar, but still, destination “importance” seems smaller than origin. Given that a significant association was found for short-range upward mobility, I would have guessed that destination would have had a slightly stronger weight. Some interpretation of these specific findings would be helpful.

### In a model, where no social mobility parameters are included, the coefficients for origin and destination capture the corresponding direct and indirect effects via mobility. When the mobility effects are captured, the relative importance of origin characteristics changes as it then only accounts for direct origin effects. Similarly, we may look at the destination’s importance and observe that its relative importance decreases once mobility effects are accounted for. This change of the relative weight importance is common in past research. For instance, see two recent studies published in Social Science and Medicine: 

Gugushvili, Alexi, Yizhang Zhao, and Erzsébet Bukodi. 2019. “‘Falling from Grace’ and ‘Rising from Rags’: Intergenerational Educational Mobility and Depressive Symptoms.” Social Science & Medicine 222:294–304. 

Steiber, Nadia. 2019. “Intergenerational Educational Mobility and Health Satisfaction across the Life Course: Does the Long Arm of Childhood Conditions Only Become Visible Later in Life?” Social Science & Medicine 242:112603. 

4. In Table 4, authors observed no interaction between race and mobility on AL. This finding would seem to contrast with the findings of Gaydosh et al., 2017, “College completion predicts lower depression but higher metabolic syndrome among disadvantaged minorities in young adulthood”, Proceedings of The National Academy of Sciences, 115(1), 109-114, who, also utilizing Add Health data, observed some physical health costs associated with upward social mobility among Black and Hispanic adults. Of course, the methods utilized in the two studies are different, but interpretation of findings from the current study considering these previous findings in Add Health are needed.

### Thank you very much for this point. We now explicitly refer to this study in the discussion section when describing the main findings of our study. In the second paragraph on page 24, you can find the following text: 

“Our finding contrast with results from past research using the same dataset, suggesting that selected minority groups (Blacks and Hispanics) experience higher metabolic syndrome after college completion [56]. Three main methodological aspects are likely to explain this difference. First, our research strategy is focused on disentangling mobility effect from origin and destination effects, based on SEP derived from educational and occupational attainment, and using DRM approach, while Gaydosh et al. rely only on educational mobility and use conventional Poisson regressions. Second, health measures also differ noticeably as Gaydosh et al. rely on metabolic syndrome based on blood pressure, glycosylated hemoglobin, body to waist ratio and cholesterol, while AL measure used in our study in addition to components such as blood pressure and cholesterol includes other biomarkers including CRP and BMI. Third, Gaydosh et al. use the restricted-full sample, while we use the public version of Add Health. These differences make any direct comparison across the two studies difficult.”

---

## [Decision Letter · Decision Letter 1]

28 Jun 2021

Socioeconomic Position, Social Mobility, and Health Selection Effects on Allostatic Load in the United States

PONE-D-21-03406R1

Dear Dr. Gugushvili,

We’re pleased to inform you that your manuscript has been judged scientifically suitable for publication and will be formally accepted for publication once it meets all outstanding technical requirements.

Kind regards,

Jennifer Morozink Boylan

Academic Editor

PLOS ONE

Additional Editor Comments (optional):

I support the authors regarding their preference to refer to their outcome as allostatic load.

Reviewers' comments:

Reviewer's Responses to Questions

**Comments to the Author**

1. If the authors have adequately addressed your comments raised in a previous round of review and you feel that this manuscript is now acceptable for publication, you may indicate that here to bypass the “Comments to the Author” section, enter your conflict of interest statement in the “Confidential to Editor” section, and submit your "Accept" recommendation.

Reviewer #1: All comments have been addressed

Reviewer #2: All comments have been addressed

Reviewer #3: All comments have been addressed

2. Is the manuscript technically sound, and do the data support the conclusions?

Reviewer #1: (No Response)

Reviewer #2: Yes

Reviewer #3: (No Response)

3. Has the statistical analysis been performed appropriately and rigorously? 

Reviewer #1: (No Response)

Reviewer #2: I Don't Know

Reviewer #3: (No Response)

4. Have the authors made all data underlying the findings in their manuscript fully available?

Reviewer #1: (No Response)

Reviewer #2: Yes

Reviewer #3: (No Response)

5. Is the manuscript presented in an intelligible fashion and written in standard English?

Reviewer #1: (No Response)

Reviewer #2: Yes

Reviewer #3: (No Response)

6. Review Comments to the Author

Reviewer #1: The authors have done a good job addressing the critiques raised by myself and the other two reviewers. I have no further major comments

Reviewer #2: (No Response)

Reviewer #3: (No Response)

7. PLOS authors have the option to publish the peer review history of their article (what does this mean?). If published, this will include your full peer review and any attached files.

Reviewer #1: No

Reviewer #2: No

Reviewer #3: No

---

## [Editor Report · Acceptance letter]

14 Jul 2021

PONE-D-21-03406R1 

Socioeconomic Position, Social Mobility, and Health Selection Effects on Allostatic Load in the United States 

Dear Dr. Gugushvili:

I'm pleased to inform you that your manuscript has been deemed suitable for publication in PLOS ONE. Congratulations! Your manuscript is now with our production department. 

Kind regards, 

on behalf of

Dr. Jennifer Morozink Boylan 

Academic Editor

PLOS ONE